# Dior-CVAE: Pre-trained Language Models and Diffusion Priors for Variational Dialog Generation

**Tianyu Yang** and **Thy Thy Tran** and **Iryna Gurevych**
Ubiquitous Knowledge Processing Lab (UKP Lab),
Department of Computer Science and Hessian Center for AI (hessian.AI),
Technical University of Darmstadt
www.ukp.tu-darmstadt.de

## Abstract

Current variational dialog models have employed pre-trained language models (PLMs) to parameterize the likelihood and posterior distributions. However, the Gaussian assumption made on the prior distribution is incompatible with these distributions, thus restricting the diversity of generated responses. These models also suffer from posterior collapse, i.e., the decoder tends to ignore latent variables and directly access information captured in the encoder through the cross-attention mechanism.

In this work, we propose Dior-CVAE, a hierarchical conditional variational autoencoder (CVAE) with diffusion priors to address these challenges. We employ a diffusion model to increase the complexity of the prior distribution and its compatibility with the distributions produced by a PLM. Also, we propose memory dropout to the cross-attention mechanism, which actively encourages the use of latent variables for response generation. Our method requires parameters that are comparable to those of previous studies while maintaining comparable inference time, despite the integration of the diffusion model. Overall, experiments across two commonly used open-domain dialog datasets show that our method can generate more diverse responses even without large-scale dialog pre-training. Code is available at https://github.com/UKPLab/dior-cvae.

## 1 Introduction

Due to the open nature of open-domain dialogs, i.e., their diverse topics and the lack of specific goals, a dialog context can be followed by multiple responses, presenting a *one-to-many* complex relationship (Csáky et al., 2019). This relationship usually poses a significant challenge to sequence-to-sequence dialog generation models that are inherently deterministic, i.e., can not produce different responses given the same dialog. Although different decoding strategies such as nucleus sampling (Holtzman et al., 2020) have been introduced

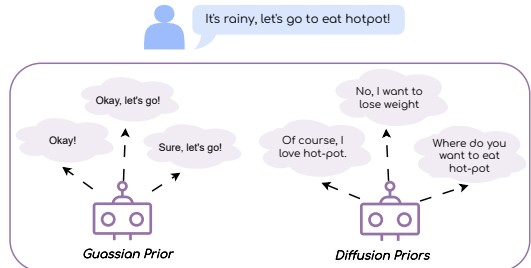

Figure 1: Limitation of the isotropic Gaussian prior distribution, i.e., generating multiple responses with similar meaning in different text forms, compared with the responses having more specific and diverse meaning offered by a diffusion model.

to bring stochasticity, these strategies mostly perform on the token level and thus might harm the fluency of the generated responses.

Conditional variational autoencoders (CVAEs) (Sohn et al., 2015) have been used to bring diversity (Zhao et al., 2017; Shen et al., 2017; Serban et al., 2017a; Chen et al., 2018, 2022; Sun et al., 2021, 2023). CVAEs draw latent variables from an assumed prior distribution conditioned on the dialog context and use these latent variables to guide the generative process. These latent variables often capture potential dialog topics, implicit conversational intents, or different styles of responses (Zhao et al., 2017).

One main challenge occurs due to the simple prior distribution, which is assumed to be the isotropic Gaussian distribution. The Gaussian assumption is oversimplified compared to the complex relationship between a dialog context and its potential responses. The Gaussian distribution is also incompatible with the expressive likelihood and posterior distributions, which are parameterized by pre-trained language models. The oversimplification and incompatibility consequently restrict the generated responses to a relatively small region of the latent space (Chen et al., 2019; Gu et al., 2019). In other words, the generated re-

sponses could be different in textual form but not in topic or intent (Fig. 1). Several studies introduce more complex prior distributions by using a neural network (NN) to sample implicit latent representations (Fang et al., 2019), or by using normalizing flows (Luo and Chien, 2021). While diffusion models have been shown to provide better priors than Gaussians and normalizing flows (Vahdat et al., 2021), they have not been used to parameterize the prior distribution for variational dialog generation.

Another major challenge of CVAEs is the well-known posterior collapse problem (Bowman et al., 2016), especially when incorporating the PLMs based on the Transformer encoder-decoder architecture (Vaswani et al., 2017). Latent variables can be easily neglected by the expressive decoder (Bowman et al., 2016) or bypassed by the cross-attention mechanism between the encoder and decoder (Bahuleyan et al., 2018). Previous studies attempt to mitigate this problem by weakening the decoder (Semeniuta et al., 2017; Zhao et al., 2017) or controlling the weight of the Kullback–Leibler (KL) divergence term (Fu et al., 2019; Li et al., 2019). Forcing the entanglement of latent variables in the decoding process has also been proposed to address the problem (Hu et al., 2022b). Different from these methods, several dropout methods have been proposed to address posterior collapse, without the need for additional training parameters (Srivastava et al., 2014; Iyyer et al., 2015; Miladinovic et al., 2022).

In this work, we propose **Dior-CVAE**, which employs a **di**ffusion model to parameterize the pri**or** distribution of a hierarchical **CVAE** model. The *diffusion model* can provide a more expressive distribution than the typical isotropic Gaussian (Ho et al., 2020). Meanwhile, the proposed model uses a Transformer-based encoder-decoder PLM to compute the posterior and likelihood distributions and derive the hierarchical latent variables. To alleviate the posterior collapse problem in Transformer-based CVAEs, we introduce *memory dropout* into the cross-attention mechanism of the decoder, which strengthens the role of latent variables in dialog response generation. Our method necessitates comparable parameters compared to prior studies and maintains comparable inference time despite of the additional diffusion model. Extensive experiments on the `DailyDialog` and `PersonaChat` datasets show better performance of our model over existing response generation methods without large-scale dialog pre-training. Our human evaluation further validates that the proposed model can generate more diverse responses with high quality.

## 2 Problem Statement and Background

### 2.1 Dialog Response Generation

Dialog response generation (DRG) aims at generating a response given a dialog context. The dialog context $c$ consists of a sequence of $N$ tokens $\mathbf{c} = [c]_1^N$, which is the concatenation of the history utterances separated by a special token (``). The response $r$ consists of $K$ tokens, $\mathbf{r} = [r]_1^K$.

### 2.2 Conditional Variational Autoencoders

Conditional Variational Autoencoders (CVAEs) learn a conditional generative model by introducing the latent variables in the form of $p(\mathbf{r}, \mathbf{z}|\mathbf{c}) = p_\psi(\mathbf{z}|\mathbf{c}) \ p_\theta(\mathbf{r}|\mathbf{z}, \mathbf{c})$ where $p_\psi(\mathbf{z}|\mathbf{c})$ is the prior distribution of the latent variable $\mathbf{z}$ given the dialog context $\mathbf{c}$ and $p_\theta(\mathbf{r}|\mathbf{z}, \mathbf{c})$ is the likelihood or decoder that generates a response $\mathbf{r}$ given latent variables $\mathbf{z}$ and the dialog context $\mathbf{c}$. Since the true posterior $p(\mathbf{z}|\mathbf{r}, \mathbf{c})$ is intractable, the generative model is often trained with an approximated posterior distribution or encoder $q_\phi(\mathbf{z}|\mathbf{r}, \mathbf{c})$. To approximate more dynamic distributions, CVAEs often use neural networks (NNs) to parameterize the prior, posterior, and likelihood distributions by $\psi$, $\phi$ and $\theta$ respectively.

**Training.** CVAEs are trained to maximize the Evidence Lower BOund (ELBO), i.e., minimize the upper bound of negative log-likelihood. The CVAE loss ($\mathcal{L}_{\text{CVAE}}$) consists of a reconstruction loss ($\mathcal{L}_{\text{RC}}$) and the Kullback-Leibler divergence ($\mathcal{L}_{\text{KL}}$). The reconstruction loss corresponds to the cross entropy between the expected and generated response. The KL divergence aligns the posterior distribution $q_\phi(\mathbf{z}|\mathbf{r}, \mathbf{c})$ with the prior $p_\psi(\mathbf{z}|\mathbf{c})$.

$$\begin{aligned}
\mathcal{L}_{\text{CVAE}} &= \mathcal{L}_{\text{RC}} + \mathcal{L}_{\text{KL}} \\
&= \mathbb{E}[-\log p_\theta(\mathbf{r}|\mathbf{z}, \mathbf{c})] \\
&\quad + \text{KL}(q_\phi(\mathbf{z}|\mathbf{r}, \mathbf{c}) \,||\, p_\psi(\mathbf{z}|\mathbf{c})).
\end{aligned} \tag{1}$$

CVAEs have shown great potential to improve the diversity of generated responses with the latent variables $\mathbf{z}$, which can represent the underlying factors such as topics, intentions, and styles associated with different responses (Zhao et al., 2017).

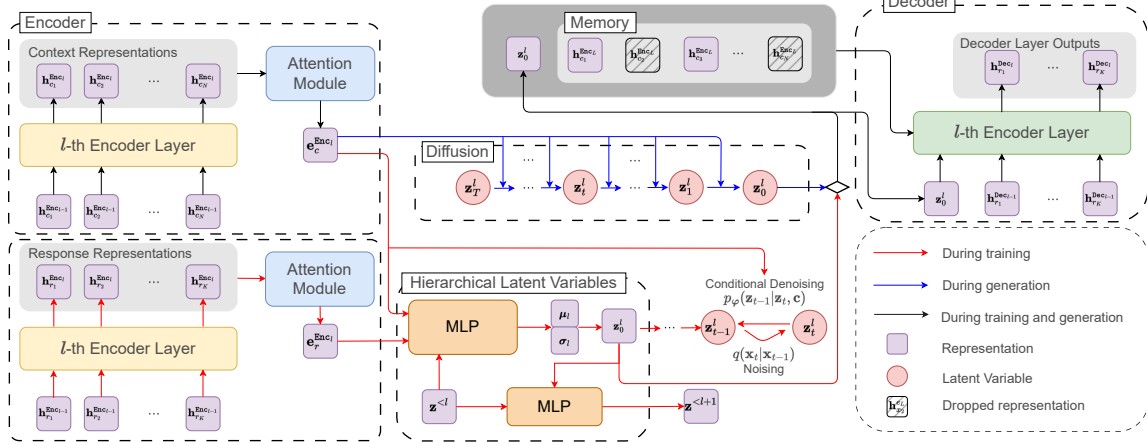

Figure 2: Our Dior-CVAE model architecture.

## 2.3 Diffusion Models

Given an observation of data $\mathbf{x}_0$, different from CVAEs, diffusion models (Ho et al., 2020) learn the data distribution $p(\mathbf{x}_0)$ by reversing a diffusion process. The diffusion (forward) process is a Markov chain that corrupts the sampled data $\mathbf{x}_0$ by gradually adding random noise to it:

$$q(\mathbf{x}_t|\mathbf{x}_{t-1}) = \mathcal{N}(\sqrt{1-\beta_t}\mathbf{x}_{t-1}, \beta_t\mathbf{I}) \quad (2)$$

where $\beta_{1:T}$ are the pre-defined noise variances, $\beta_t \in (0,1)$ at time step $t$. When $\beta_t \to T$, the data distribution will be corrupted to $\mathcal{N}(0,\mathbf{I})$. By defining $\alpha_t = \prod_{i=1}^{t}(1-\beta_i)$, we can directly get $\mathbf{x}_t$ by adding noise to the input as follows:

$$q(\mathbf{x}_t|\mathbf{x}_0) = \mathcal{N}(\sqrt{\alpha_t}\mathbf{x}_0, (1-\alpha_t)\mathbf{I}) \quad (3)$$

where $\alpha_t \in (0,1)$.

Given access to the original data $\mathbf{x}_0$, the forward process can be inverted analytically

$$p(\mathbf{x}_{t-1}|\mathbf{x}_t, \mathbf{x}_0) = \mathcal{N}(f_t(\mathbf{x}_t, \mathbf{x}_0), \sigma_t^2\mathbf{I}) \quad (4)$$

where $\sigma_t$ can be derived from $\beta_t$ (Ho et al., 2020), $f_t(\mathbf{x}_t, \mathbf{x}_0)$ has a closed form (Ho et al., 2020) parameterized by $t$. However, since the original data $\mathbf{x}_0$ is not available in the actual generation process, (i.e., the response is supposed to be generated), we can not directly use Eq. (4) to sample data. We thus approximate $f_t(\cdot)$ using an NN with the parameter $\varphi$, namely *denoising network*. The training objective of the denoising network can be defined as:

$$\mathbb{E}_{t,\mathbf{x}_0,\mathbf{x}_t}\left[\frac{1}{2\sigma_t^2}||f_\varphi(\mathbf{x}_t, t) - \mathbf{x}_0||\right] \quad (5)$$

where $t \sim \text{Uniform}(\{1, \cdots, T\})$, $\mathbf{x}_t \sim q(\mathbf{x}_t|\mathbf{x}_0)$.

For inference, we can use the trained denoising network $f_\varphi(\mathbf{x}_t, t)$ to build the usable inversion $p_\varphi(\mathbf{x}_{t-1}|\mathbf{x}_t) \approx p(\mathbf{x}_{t-1}|\mathbf{x}_t, f_\varphi(\mathbf{x}_t, t))$, referring to

lines 16-17 of Alg. 1, and get new high-quality data by sampling from it iteratively.

## 3 Our Method – Dior-CVAE

We present Dior-CVAE, a hierarchical CVAE model based on an encoder-decoder Transformer, with four improvements (Fig. 2). First, we enhance the computation of hierarchical latent variables with attention mechanism (§3.1). These variables are then infused into the decoder via self- and cross-attention (§3.2). We then introduce memory dropout during training to alleviate posterior collapse, a well-known problem in CVAEs (§3.3). Most importantly, we parameterize the prior distribution using a diffusion model for more flexible and compatible representations than an isotropic Gaussian (§3.4). Finally, we introduce the objective for the end-to-end training and describe the training and inference process (§3.5).

### 3.1 Hierarchical Latent Variables

Hierarchical CVAEs (Sønderby et al., 2016; Klushyn et al., 2019; Vahdat and Kautz, 2020; Child, 2021) increase the expressiveness of the approximated prior and posterior by splitting the latent variables into $L$ groups $\mathbf{z} = \{\mathbf{z}^1, \cdots, \mathbf{z}^L\}$. The prior and approximated posterior of the latent variables $\mathbf{z}$ can be factorized as:

$$p_\psi(\mathbf{z}|\mathbf{c}) = \prod_{l=1}^{L} p_{\psi^l}(\mathbf{z}^l|\mathbf{z}^{<l}, \mathbf{c}) \quad (6)$$

$$q_\phi(\mathbf{z}|\mathbf{r}, \mathbf{c}) = \prod_{l=1}^{L} q_{\phi^l}(\mathbf{z}^l|\mathbf{z}^{<l}, \mathbf{r}, \mathbf{c}). \quad (7)$$

where $\psi^l$, $\phi^l$ denote parameters of the $l$-th layer. When $l = 1$, $p_{\psi^1}(\mathbf{z}^1|\mathbf{z}^{<1}, \mathbf{c}) = p_{\psi^1}(\mathbf{z}^1|\mathbf{c})$. The same applies for $q_{\phi^1}(\mathbf{z}^1|\mathbf{z}^{<1}, \mathbf{r}, \mathbf{c}) = q_{\phi^l}(\mathbf{z}^1|\mathbf{r}, \mathbf{c})$. The detailed building process of the posterior

(Eq. (7)) and the prior (Eq. (6)) distribution will be introduced in the following content and the §3.4, respectively.

In this work, we employ an encoder-decoder PLM with $L$ encoder (Enc) and $L$ decoder (Dec) layers to build the hierarchical CVAE. Each layer corresponds to a group of latent variables. We denote the hidden output by the $l$-th encoder layer as $\mathbf{H}_c^{\text{Enc}_l} = \text{Enc}_l(\mathbf{H}_c^{\text{Enc}_{l-1}}) \in \mathbb{R}^{N \times d}$. We construct the mean and variance of the approximated posterior $q_{\phi^l}(\mathbf{z}^l | \mathbf{z}^{<l}, \mathbf{r}, \mathbf{c})$ as follows:

$$\begin{bmatrix} \boldsymbol{\mu}_{q_\phi}^l \\ \log(\boldsymbol{\sigma}_{q_\phi}^l) \end{bmatrix} = \text{FNN}(\begin{bmatrix} \mathbf{z}^{<l} \\ \mathbf{e}_c^{\text{Enc}_l} \\ \mathbf{e}_r^{\text{Enc}_l} \end{bmatrix}) \qquad (8)$$

where FNN refers to a fully-connected feed forward NN with tanh as the activation function, $[\cdot]$ denotes the concatenation of the representations. The mean and variance can be used to sample latent variables $\mathbf{z}^l$ using the re-parameterization trick (Kingma et al., 2021) to enable back-propagation. The inputs to the FNN are computed as follows.

We aggregate information from the latent variables of the lower layers to get the $\mathbf{z}^{<l}$ in the Eq. (8):

$$\mathbf{z}^{<l} = \text{FNN}(\begin{bmatrix} \text{Linear}(\mathbf{z}^{<l-1}) \\ \text{Linear}(\mathbf{z}^{l-1}) \end{bmatrix}) \qquad (9)$$

where Linear denotes a fully-connected NN without activation function.

We construct the representation $\mathbf{e}_c^{\text{Enc}_l}$ in the Eq. (8) from each encoder layer by attending over all outputs of that layer:

$$\mathbf{e}_c^{\text{Enc}_l} = \text{Att}(\mathbf{H}_c^{\text{Enc}_l}) \qquad (10)$$

where Att refers to the attention mechanism (Yang et al., 2016). We can obtain the representation $\mathbf{e}_r^{\text{Enc}_l}$ in the Eq. (8) following the same method.

### 3.2 Hierarchical Latent Variables Infusion

We prepend the latent variables as prefix embeddings to the input of the self-attention mechanism in each decoder layer, following previous work (Chen et al., 2022):

$$\text{SelfAtt}^l([\text{Linear}(\mathbf{z}^l), \mathbf{H}^{\text{Dec}_{l-1}}]) \qquad (11)$$

The latent variables can then be iteratively infused to generate the subsequent tokens. Differently, the cross attention takes the output of the final encoder layer $\mathbf{H}_c^{\text{Enc}_L}$ as **memory**. Cross-attention has shown its importance in the Transformer model, resulting in more quality degradation when pruned (Voita et al., 2019). We thus prepend the latent variables to the (encoder) memory, which is passed to the cross-attention mechanism.

$$\text{XAtt}^l([\text{Linear}(\mathbf{z}^l), \mathbf{H}_c^{\text{Enc}_L}]) \qquad (12)$$

Intuitively, the latent variables can serve as additional memory for the decoder. The model facilitates the next token generation with the deep infusion of latent variables. However, latent variables and (encoder) memory theoretically contain overlapping information. The latent variables may be ignored during the generation process, causing posterior collapse. This drawback can be mitigated by the *memory dropout* introduced in §3.3.

### 3.3 Memory Dropout

In this work, we propose memory dropout to address posterior collapse problem. The memory dropout aims at encouraging the use of the latent variables in the decoder. In particular, we apply random dropout to the hidden state: $\mathbf{h}_{c_i}^{\text{Enc}_L}$ of the memory where $i \in [1, N]$ while keeping the latent variables. Subsequently, the input of the cross-attention in each decoder layer becomes:

$$\text{XAtt}^l([\text{Linear}(\mathbf{z}^l), \text{memdrop}(\mathbf{H}_c^{\text{Enc}_L})]) \qquad (13)$$

where $\text{memdrop}(\cdot)$ denotes the memory dropout operation with a certain probability. Concretely, this is done by randomly masking out some hidden states from the memory of the cross-attention mechanism. Compared with previous methods, our memory dropout does not introduce any additional trainable parameters (Miladinovic et al., 2022).

### 3.4 Diffusion Prior

We parameterize the prior distribution $p_\psi(\mathbf{z}|\mathbf{c})$ defined in the Eq. (6) with a diffusion model to improve its complexity. The conditional information, dialog context $\mathbf{c}$, can be introduced as an additional input for the denoising network, as $f_\varphi(\mathbf{x}_t, t, \mathbf{c})$. During training, latent variables sampled from the approximated posterior distribution are used as the input data $\mathbf{x}_0 := \mathbf{z}$ (Eq. (7)). The diffusion model is trained to imitate the posterior distribution. During inference, the denoising network conditioned on the dialog context is used to sample representations of latent variables.

Specifically, to condition on the latent variables in lower layers as done in the inference of the posterior distribution (§3.1) and the dialog context $c$, we concatenate the latent variables sampled from the posterior distribution and conditional representa-

**Algorithm 1** Dior-CVAE Inference

**Input** Dialog context $\mathbf{c}$, # timestep $T$, noise schedule $[\alpha_t]_1^T$, sampling hyperparameter $[\sigma_t]_1^T$,
**Model** Denoising model $f_{\boldsymbol{\varphi}}(\cdot)$
**Output** Response $\mathbf{r}$
1:  $\mathbf{H}_c^{\mathsf{Enc}_0} \leftarrow \mathbf{c}$          ▷ Embed the tokens
2:  **for** $l = 1, ..., L$ **do**
3:      $\mathbf{H}_c^{\mathsf{Enc}_l} = \mathsf{Enc}_l(\mathbf{H}_c^{\mathsf{Enc}_{l-1}})$
4:      Get $\mathbf{e}_c^{\mathsf{Enc}_l}$ through $\mathbf{H}_c^{\mathsf{Enc}_l}$ according Eq. (10)
5:  **end for**
6:  Get $\mathbf{e}_c$ by concatenating $[\mathbf{e}_c^{\mathsf{Enc}_l}]_1^L$ according Eq. (14).
7:  $\mathbf{z}_T \in \mathbb{R}^d \sim \mathcal{N}(0, \mathbf{I})$         ▷ Sample noise
8:  **for** $t = T, ..., 1$ **do**
9:      $\mathbf{z} = (1 + w) * f_{\boldsymbol{\varphi}}(\mathbf{e}_c, t, \mathbf{z}_t) - w * f_{\boldsymbol{\varphi}}(\mathbf{0}, t, \mathbf{z}_t)$
10:     **if** $t == 1$ **then**
11:        **return** $\mathbf{z}$
12:     **end if**
13:     $\boldsymbol{\epsilon} \in \mathbb{R}^d \sim \mathcal{N}(0, \mathbf{I})$
14:     $\tilde{\boldsymbol{\epsilon}}_t = \frac{\mathbf{z}_t - \sqrt{\alpha_t}\mathbf{z}}{\sqrt{1 - \alpha_t}}$
15:     $\mathbf{z}_{t-1} = \sqrt{\alpha_{t-1}}\mathbf{z} + \sqrt{1 - \alpha_{t-1} - \sigma_t^2}\tilde{\boldsymbol{\epsilon}}_t + \sigma_t\boldsymbol{\epsilon}$
16: **end for**
17: Split $\mathbf{z}$ into $[\mathbf{z}^l]_1^L \in \mathbb{R}^{d/L}$.
18: $\mathbf{r} = \mathsf{Dec}([\mathbf{z}^l]_1^L, \mathbf{H}_c^{\mathsf{Enc}_L})$

---

**Algorithm 2** Training Dior-CVAE

**Input** Dataset $\mathcal{D}$, # timestep $T$, noise schedule $[\alpha_t]_1^T$
**Model** Denoising model $f_{\boldsymbol{\varphi}}(\cdot)$
1:  **while** not converged **do**
2:      $(\mathbf{c}, \mathbf{r}) \sim \mathcal{D}, \mathbf{z}^{<1} = \mathbf{0}$     ▷ Sample data, init latent
3:      **for** $l = 1, ..., L$ **do**
4:         $\mathbf{H}_c^{\mathsf{Enc}_l} = \mathsf{Enc}_l(\mathbf{H}_c^{\mathsf{Enc}_{l-1}})$    ▷ Context embed.
5:         $\mathbf{e}_c^{\mathsf{Enc}_l} = \mathsf{Att}(\mathbf{H}_c^{\mathsf{Enc}_l})$      ▷ Eq. (10)
6:         $\mathbf{H}_r^{\mathsf{Enc}_l} = \mathsf{Enc}_l(\mathbf{H}_r^{\mathsf{Enc}_{l-1}})$    ▷ Response embed.
7:         $\mathbf{e}_r^{\mathsf{Enc}_l} = \mathsf{Att}(\mathbf{H}_r^{\mathsf{Enc}_l})$      ▷ Eq. (10)
8:         Get $\boldsymbol{\mu}_{q_\phi}^l, \log(\boldsymbol{\sigma}_{q_\phi}^l)$ from Eq. (8)
9:         $\mathbf{z}^l \sim \mathcal{N}(\boldsymbol{\mu}_{q_\phi}^l, \boldsymbol{\sigma}_{q_\phi}^l)$
10:     **end for**
11:     $\mathbf{z} = [\mathbf{z}^1 \cdots \mathbf{z}^L]^\top$          ▷ Eq. (14)
12:     $t \sim \mathrm{Uniform}(\{1, ..., T\})$    ▷ Sample timestep
13:     $\mathbf{z}_t \sim \mathcal{N}(\sqrt{\alpha_t}\mathbf{z}, (1 - \alpha_t)\mathbf{I})$   ▷ Sample $\mathbf{z}$ at time $t$
14:     $\omega \sim \mathrm{Uniform}([0, 1])$
15:     **if** $\omega < \eta$ **then**
16:        $\mathbf{e}_c = \mathbf{0}$
17:     **else**
18:        $\mathbf{e}_c = [\mathbf{e}_c^{\mathsf{Enc}_1} \cdots \mathbf{e}_c^{\mathsf{Enc}_L}]^\top$    ▷ Eq. (14)
19:     **end if**
20:     $\mathcal{L} = \mathcal{L}_{\mathtt{RC}} + \mathcal{L}_{\mathtt{neg-xent}} + \mathcal{L}_{\mathtt{xent}}$    ▷ Eq. (16)
21:     Calculate gradients and update parameters
22: **end while**

---

tions from all layers following the below equation:

$$\mathbf{z} = [\mathbf{z}^1 \cdots \mathbf{z}^L]^\top$$
$$\mathbf{e}_c = [\mathbf{e}_c^{\mathsf{Enc}_1} \cdots \mathbf{e}_c^{\mathsf{Enc}_L}]^\top \quad (14)$$

The sinusoidal position embeddings (Vaswani et al., 2017) are adopted to represent timestep $t$ to get the time embedding $\mathrm{pe}(t)$, which is first added to the conditional representation and then concatenated with the noisy latent variables $\mathbf{z}_t$ to form the input of the denoising network. Thus, the denoising network can be defined as:

$$f_{\boldsymbol{\varphi}}(\mathbf{e}_c, t, \mathbf{z}_t) = \mathtt{FNN}\left(\mathtt{Linear}\left(\begin{bmatrix} \mathrm{pe}(t) + \mathbf{e}_c \\ \mathbf{z}_t \end{bmatrix}\right)\right) \quad (15)$$

The general diffusion model described in §2.3 can only model the unconditional distribution. To obtain a conditional diffusion model, we follow the the classifier-free guidance paradigm (Ho and Salimans, 2021), where we train a conditional and an unconditional diffusion model simultaneously by replacing the conditional representation $\mathbf{e}_c$ by a zero vector with a probability $\eta$ during training. During inference, the output interpolation of these two models with weight $w$ is used as the final prior representation, referring to line 9 of Alg. 1.

### 3.5 End-to-end Training

As mentioned in §2.2, the training objective of CVAEs consists of the reconstruction and the KL divergence losses. To learn also the latent diffusion prior simultaneously, we follow Vahdat

et al. (2021) to decompose the KL loss into its negative entropy ($\mathcal{L}_{\mathtt{neg-xent}}$) and cross-entropy ($\mathcal{L}_{\mathtt{xent}}$) losses. The reconstruction and the negative entropy terms can be calculated using the reparameterization trick (Kingma and Welling, 2014). The cross-entropy term can be further expressed with the regression loss ($\mathcal{L}_{\mathtt{reg}}$) of the denoising diffusion model defined in Eq. (5) (Vahdat et al., 2021). The final loss of Dior-CVAE is as follows:

$$\begin{aligned} \mathcal{L} &= \mathcal{L}_{\mathtt{RC}} + \mathcal{L}_{\mathtt{KL}} \\ &= \mathcal{L}_{\mathtt{RC}} + \mathcal{L}_{\mathtt{neg-xent}} + \mathcal{L}_{\mathtt{xent}} \\ &= \mathcal{L}_{\mathtt{RC}} + \mathcal{L}_{\mathtt{neg-xent}} + \mathcal{L}_{\mathtt{reg}} \\ &= \mathbb{E}[-\log p_{\boldsymbol{\theta}}(\mathbf{r}|\mathbf{z}, \mathbf{c})] \quad (16) \\ &\quad + \mathbb{E}[\log q_{\phi}(\mathbf{z}|\mathbf{r}, \mathbf{c})] \\ &\quad + \mathbb{E}\left[\frac{1}{2\sigma_t^2}||f_{\boldsymbol{\varphi}}(\mathbf{e}_c, t, \mathbf{z}_t,) - \mathbf{z}||\right] \end{aligned}$$

**Training (Alg. 2).** The latent variables are sampled from the approximate posterior distribution of each layer where the parameters of the distribution are calculated through the layer-wise conditional representation and reference response representation. In addition to being fed into the decoder to generate the target response, the latent variables are also used as the input data $\mathbf{x}_0$ in the diffusion model to train the diffusion model to imitate the posterior distribution.

**Inference (Alg. 1).** Specifically, to generate the response for a given dialog context, we first encode the dialog context and get the conditional representation from each layer of the encoder. The representations are then concatenated as one of the inputs to the denoising network. Starting from the final step T, we first sample the latent variables $\mathbf{z}_T$ from the standard Gaussian distribution. Then we iteratively denoise the latent variables conditioned on the concatenated conditional representations using the denoising network until step 1 when we get the latent variables $\mathbf{z}$. We split $\mathbf{z}$ into $L$ parts, resulting in $\mathbf{z}^1, \cdots, \mathbf{z}^L$ and feed them into each layer of the decoder along with the memory to generate the response.

## 4 Experiments

This section gives a brief overview of our experimental settings. We refer to appendices A to C for a full set of hyperparameters, data statistics, and formal definitions of metrics, respectively.

**Implementation.** Our model was developed using the OpenNMT-py library (Klein et al., 2017). We employed BART (Lewis et al., 2020) as the backbone PLM, with the max sequence length set as 1024 and 50 diffusion steps.

**Datasets & Metrics.** We trained and evaluated the proposed model on the DailyDialog (Li et al., 2017) and Persona-Chat (Zhang et al., 2018) datasets. DailyDialog is a collection of English dialogs about daily life, while Persona-Chat additionally includes personas of the speakers. We follow previous work in reporting the lexical similarity of the references and generated responses using BLEU-1/2 (Papineni et al., 2002) and the lexical diversity calculated by Distinct-1/2 (Li et al., 2016) which computes the ratio of distinct $n$-grams in the generated responses.

In addition, we employ Entropy-1/2/3 (Malashina, 2021) to measure meaningful information in the generated responses. Since lexical-overlapping metrics have shown great limitations for text generation, we then employ model-based metrics for better evaluation, including BERTScore (BTS) (Sun et al., 2022) and FED (Mehri and Eskenazi, 2020). BERTScore (BTS) measures the *semantic* similarity of the reference and generated responses (Devlin et al., 2019). We also employ FED (Mehri and Eskenazi, 2020) which measures 18 fine-grained qualities of

the generated response, including the relevancy, coherency, diversity, and understandability.

**Analysis** For diversity analysis of the generated responses, we sample 100 dialogs in the intersection of DailyDialog and DailyDialog++ (Sai et al., 2020), which has multiple references for each dialog, namely DailyDialog-100.

**Baselines.** We compare Dior-CVAE with the state-of-the-art models for variational dialog generation. Overall, the baselines are mostly the Transformer-based models pre-trained on the large-scale corpus. One of the critical differences between these models is whether they are pre-trained on a large-scale dialog dataset. More details about the baselines can be seen in the Appx. D.

## 5 Experimental Results

### 5.1 Main Results

Tab. 1 presents the main evaluation results on the test sets of DailyDialog and Persona-chat. On DailyDialog, our model surpasses all baselines by a large margin for all metrics, while getting comparable performance as the models pre-trained on large-scale dialog data such as PLATO and DialogVED. The higher performance demonstrates the expressive representation capability of the diffusion priors in combination with the PLMs to generate high-quality responses. The proposed model can bring the generative distribution closer to the true dialog distribution. Regarding Persona-chat, once again, Dior-CVAE, with fewer parameters, mostly achieves better diversity than SoTA models. Compared to models with large-scale dialog pre-training, the results are inconclusive, with higher results than PLATO but slightly lower than DialogVED. The inconsistent results indicate the potential positive impact of dialog pre-training but investigation is required for further understanding.

**Further analysis.** We additionally report in Tab. 2 the $n$-gram Entropy scores and evaluation results of the model-based metrics. The Entropy scores show that our proposed method can generate more diverse responses compared to the large-scale dialog pre-trained model – DialogVED. BERTScore (BTS) focuses on semantic similarity brought by the cosine similarity between the contextual embeddings of a reference and a generated response. We can see a similar trend in BERTScore (BTS) on the DailyDialog dataset

| Model | Size (mil.) | DailyDialog | | | | PersonaChat | | | |
|---|---|---|---|---|---|---|---|---|---|
| | | BLEU-1 | BLEU-2 | Distinct-1 | Distinct-2 | BLEU-1 | BLEU-2 | Distinct-1 | Distinct-2 |
| | | *without dialog pre-training* | | | | | | | |
| iVAE$_{\text{MI}}^{\dagger}$ | 3.9 | 30.9 | 24.9 | 2.9 | 25.0 | 38.2 | 27.7 | 0.9 | 8.2 |
| LIC | 117 | - | - | - | - | 40.5 | 32.0 | 1.9 | 11.3 |
| DELLA (BART) † | 209 | 47.3 | 40.5 | 5.2 | 28.9 | 41.6 | 35.4 | 1.7 | 9.8 |
| Optimus$^{\dagger}$ | 227 | 41.2 | 38.5 | 4.1 | 29.7 | 42.7 | 34.3 | 1.9 | 11.7 |
| ProphetNet | 332 | 44.3 | 39.2 | 3.9 | 21.1 | **46.6** | **39.1** | 1.3 | 7.5 |
| DRESS | 406 | - | - | 5.4 | 29.1 | - | - | - | - |
| MVP+S$^{\dagger}$ | 406 | 45.7 | 42.9 | 5.1 | 27.1 | 43.4 | 35.8 | 2.0 | 11.1 |
| Dior-CVAE-sampling (ours) | 237 | 50.3 | 46.7 | **7.0** | **35.1** | 42.6 | 36.1 | **2.8** | **26.5** |
| Dior-CVAE-beam (ours) | 237 | **52.0** | **47.8** | 6.3 | 31.1 | 44.1 | 37.2 | 1.9 | 13.1 |
| | | *with large-scale dialog pre-training* | | | | | | | |
| PLATO | 115 | 39.7 | 31.1 | 5.4 | 29.1 | 40.6 | 31.5 | 2.1 | 12.1 |
| DialogVED-sampling | 392 | 43.1 | 37.0 | 5.8 | 37.2 | 42.8 | 35.7 | 3.2 | 27.3 |
| DialogVED-beam | 392 | 48.1 | 42.1 | 4.2 | 23.2 | 48.2 | 39.9 | 1.5 | 9.4 |

Table 1: Performance on the `DailyDialog` and `Persona-chat` test sets in comparison with the state-of-the-art. Results of the previous methods denoted by † are implemented and evaluated by us.

| Model | Diversity | | | Model-based | |
|---|---|---|---|---|---|
| | Ent-1 | Ent-2 | Ent-3 | BTS | FED |
| Dior-CVAE | **3.56** | **4.83** | **5.07** | **87.57** | **5.36** |
| DialogVED | 3.44 | 4.57 | 4.75 | 86.76 | 5.29 |

Table 2: Diversity and model-based evaluation on the test set of `DailyDialog`. Ent-$n$ indicates Entropy-$n$ where $n$ stands for $n$-gram, BTS is BERTScore (Devlin et al., 2019), FBD (Xiang et al., 2021).

| Model | BLEU-1 | BLEU-2 | Distinct-1 | Distinct-2 |
|---|---|---|---|---|
| Dior-CVAE | 57.0 | 45.0 | 7.0 | 36.3 |
| *Prior distribution* | | | | |
| w/o diffusion (with Gaussian Prior) | 49.3 | 41.1 | 4.7 | 22.1 |
| *Training* | | | | |
| w/o memory dropout | 57.3 | 45.4 | 6.7 | 34.7 |
| *Architecture* | | | | |
| w/o self-attention infusion | 56.4 | 44.4 | 6.8 | 36.0 |
| w/o cross-attention infusion | 55.2 | 43.7 | 6.8 | 35.4 |
| w/o PLM | 43.2 | 38.1 | 3.7 | 31.2 |

Table 3: Ablation results on the validation set of the `DailyDialog` dataset. w/o (without) denotes the removal of the corresponding component.

compared to that of the lexical-overlapping metrics BLEU-$n$. For both metrics, our method achieves higher scores than DialogVED. Also, the higher FED score by our model indicates the higher quality of generated responses on multiple dimensions, including the relevance, coherence, diversity, and understandability of the responses.

## 5.2 Ablation Study

We conduct an ablation analysis on the validation set of the `DailyDialog` validation set (Tab. 3). We compare Dior-CVAE with the following ablated variants: w/o diffusion: The diffusion model is removed and the prior distribution is assumed to be the isotropic Gaussian distribution; w/o memory dropout: memory dropout is removed; w/o self-attention infusion: Eq. (11) is not computed; w/o cross-attention infusion: Eq. (12) is not computed; w/o PLM: random initialization of the Transformer encoder and decoder is used instead of BART.

We observe that the diffusion prior and latent variable infusion greatly contribute to both metrics that evaluate the coherence and the diversity of the generated responses. Unlike the above two components, memory dropout mainly contributes

to diversity (Distinct-1/2) while slightly harming lexical-overlapping scores (BLEU-1/2). While memory dropout is designed to promote diversity, generated responses then can be diverse from the references, leading to a slight decrease in lexical-overlapping scores. We further generate responses by sampling different latent variables to assess the effects of memory dropout, the analysis can be found in Appx. H. We also ablate the PLM to assess whether generic large-scale pre-training is useful for dialog generation. The great performance drop after removing PLM highlights the importance of pre-training. Interestingly, the diversity remains relatively high even without using PLM, which is greatly attributed to the diffusion model.

## 5.3 Human Evaluation

Since automatic metrics for open-domain text generation may not be consistent with human perceptions (Liu et al., 2016), we also conduct a human evaluation on the `DailyDailog-100` subset with the help of three expert annotators. All annota-

| Model | Quality | | | | Diversity |
|---|---|---|---|---|---|
| | COH | INF | SAF | ENG | |
| Human | 1.908 | 1.862 | 2.000 | 1.947 | 4.831 |
| Dior-CVAE | **1.534** | **1.601** | **1.993** | **1.693** | **1.845** |
| DialogVED | 1.215 | 1.378 | 1.983 | 1.433 | 1.500 |

Table 4: Human evaluation on `DailyDialog-100` subset.

tors have an NLP background. For each dialog, we sample five responses from Dior-CVAE and DialogVED. For quality, each annotator is asked to judge the quality with regards to the following four criteria: Coherent (COH), Informative (INF), Safety (SAF), and Engagement (ENG) on a 3-point Likert scale. We describe the criteria details in Appx. E. Furthermore, we automatically mark responses that do not violate any criteria as *valid*, i.e., only a maximum of 5 generated responses are valid. For the diversity evaluation, annotators are asked to annotate the number of distinct meanings among the *valid* responses. Results of the human evaluation are reported in Tab. 4. Compared to DialogVED, our method not only generates higher quality but also more diverse responses.

### 5.4 Perplexity of Multiple References

To further verify that our proposed model can generate more diverse responses, we calculate the perplexity of multiple different responses given the same context (Fig. 3). In particular, given a dialog context, we sample one to five human references from DailyDialog-100 (x-axis). We calculate the averaged perplexity of our method and its ablation without diffusion priors (i.e., with Gaussian priors) on the sampled human references. We also compute the cosine similarity between every two reference responses for the same dialog context using BERT (set as 1 when there is only one reference).

From the cosine similarity shown by the blue curve, we can see that the human-labeled responses for the same dialog context are semantically different from each other. We can notice that the perplexity scores by the ablation without diffusion priors are significantly higher than our method. This indicates that diffusion models can better approximate multiple potential responses given a dialog context which are semantically different.

### 5.5 Analysis with Llama-2

In this section, we compare our model with Llama-2 (Touvron et al., 2023), one of the most recent SoTA Large Language Models (LLMs) with bil-

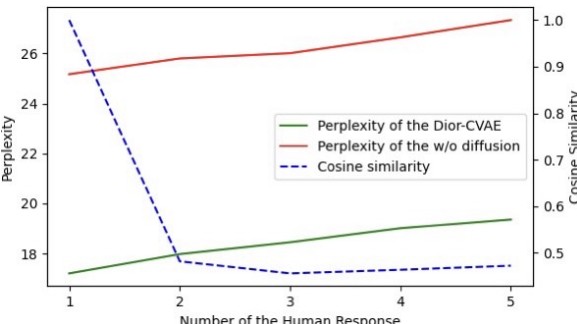

Figure 3: Perplexity of two approaches and cosine similarity of human references on the `DailyDialog-100` dataset. The decreasing trend of the cosine similarity curve demonstrates the diversity of the reference responses, and two perplexity curves show the average perplexity of the two methods, Dior-CVAE and its ablation without diffusion priors.

| Model | B-1 | B-2 | D-1 | D-2 | BTS | FED |
|---|---|---|---|---|---|---|
| ICL | 13.7 | 14.0 | **8.2** | 24.1 | 83.43 | 5.88 |
| LoRA | 39.6 | 35.4 | 7.1 | **41.1** | 85.98 | **5.91** |
| Dior-CVAE | **52.0** | **47.8** | 6.3 | 31.1 | **87.57** | 5.36 |

Table 5: LLM performance evaluated on the test set of the `DailyDialog` dataset.

lions of parameters (Brown et al., 2020; Touvron et al., 2023; Chowdhery et al., 2023). Specifically, we prompt the Llama-2 to generate responses given a dialog history through in-context learning (ICL) (Brown et al., 2020) and instruction tuning (Mishra et al., 2022). A parameter-efficient fine-tuning method (LoRA) (Hu et al., 2022a) is used for the instruction tuning.

The evaluation results are shown in Tab. 5. Dior-CVAE performs better on the BLEU-1/2 metrics while LLAMA-2 gets higher Dist-1/2 scores. This indicates that the responses generated by the LLM have the best lexical diversity. While BERTScore (BTS) focuses on semantic similarity between the reference and generated responses, our method also gets the best performance on this. The generated responses by Dior-CVAE match better with the reference responses. In contrast, LLAMA-2 gets higher FED scores, suggesting that the responses generated by LLMs may have better quality on multiple criteria. This also emphasizes the functionality of large-scale pre-training for dialog systems.

## 6 Related Work

**Variational dialog generation.** Conditional variational autoencoders (CVAE) (Kingma and Welling, 2014; Sohn et al., 2015) achieved impressive results to address the *safe and commonplace*

*response* problem in the dialogue generation task by representing dialog contexts in the latent space (Zhao et al., 2017; Shen et al., 2017; Serban et al., 2017b; Chen et al., 2018). One limitation is the oversimplified Gaussian assumption of the prior and the posterior distributions. Several studies (Serban et al., 2017a; Zhao et al., 2018; Gao et al., 2019; Cai and Cai, 2022) introduce discrete latent variables to improve the complexity of these distributions. Further studies use more advanced generative models like Generative Adversarial Network (Goodfellow et al., 2020; Gu et al., 2019; Khan et al., 2020) or Normalizing Flows (Rezende and Mohamed, 2015; Luo and Chien, 2021).

**Diffusion models for text generation.** Adapting diffusion models to natural language remains an open challenge due to the inherently discrete nature of texts. These studies can be divided into discrete and continuous. Notable work (Austin et al., 2021; Hoogeboom et al., 2021, 2022) directly defines a forward diffusion process on discrete data. Other work has adapted diffusion models in the word embedding space and presented them as an alternative to auto-regressive LMs (Li et al., 2022; Gong et al., 2023; Strudel et al., 2022). Differently, diffusion models have been studied in the latent space to complement existing PLMs (Liu et al., 2022; Lovelace et al., 2022; Yu et al., 2022), which condition on a pre-defined set of labels. In our approach, we investigate how to incorporate diffusion priors in variational dialog generation.

## 7 Conclusion & Future Work

We proposed Dior-CVAE, an approach for variational dialog generation, which incorporates a diffusion model to produce a more informative and expressive prior distribution. Our method is based on a hierarchical conditional variational autoencoder (CVAE), which derives latent variables from every encoder layer and fuses them into the corresponding decoder layers as hierarchical latent memory. A pre-trained language model, BART, is employed to estimate the posterior and likelihood distributions of the CVAE. The proposed approach approximates the one-to-many complex relationship of dialog response generation, i.e., multiple potential responses given a dialog context. The approach does not require more parameters than previous work, and the inference time remains comparable regardless of the introduction of the diffusion model. We also propose memory dropout to alleviate the posterior collapse problem in training Transformer-based CVAEs.

Our experiments across two commonly used dialog datasets show that the proposed method can generate diverse responses without relying on large-scale dialog pre-training. This work suggests the effectiveness of using a diffusion model to parameterize the prior distribution in Transformer-based CVAEs for dialog response generation. Future work on diffusion models for text generation in general should be explored given their potential.

## Limitations

One limitation of this work is the instability of the training process due to the high variance of the time step sampling operation in the diffusion model (Eq. (5)). An advanced pre-defined noise variances scheme for timestep sampling (Vahdat et al., 2021; Nichol and Dhariwal, 2021) will be explored to address this problem. Future work also aims at understanding information captured in the latent space produced by diffusion models (Khrulkov et al., 2023), towards an interpretable latent space.

This work only considers BART, a Transformer encoder-decoder architecture, as our backbone PLM. However, many recent SoTA large language models (LLMs) are decoder-only architecture, further experiments are required to use these LLMs with diffusion priors.

## Ethics Considerations

This work is fine-tuned and evaluated on two publicly available datasets that are widely used as benchmarks for open-domain dialog generation. However, due to the use of a PLM, the fine-tuned models may inherit biases from large-scale pre-training. Safety filtering and/or debiasing methods should be considered while using the proposed approach in real-world applications.

## Acknowledgement

This work has been funded by the European Union under the Horizon Europe grant No 101070351 (SERMAS) and by the German Federal Ministry of Education and Research (BMBF) under the promotional reference 13N15897 (MISRIK). We also thank our internal and anonymous reviewers for their constructive comments on this paper.

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

# A Hyperparameters

Dior-CVAE uses BART (Lewis et al., 2020) as the backbone PLM which consists of a 6-layer encoder and a 6-layer decoder. The size of hidden state is 768. The max length of the input tokens is 1024 and the max length the generated response is set as 200. To compare with the baselines, the size of latent variable is set as 64.

We use Adam optimizer (Kingma and Ba, 2015) with a learning rate of $5 \times e^{-4}$ on the Daily-Dialog dataset and $1 \times e^{-4}$ on the Persona-Chat dataset. We train the model for 500,000 steps, which takes around 78 hours in total. The learning rate schedule is set according to the study from Vaswani et al..

Warmup steps of the optimization is set as 20,000 and 40,000 on the Daily-Dialog and Persona-Chat, respectively. Besides, we utilize KL annealing tricks to mitigate the posterior problem, the weight of KL term in the ELBO increases to 1 in 20,000 steps linearly. In the computation of negative log loss, the label smoothing value is set as 0.1. We use the dynamic batching mechanism in the Open-NMT package. The batch size measured by tokens is 4096. After every 5,000 optimization steps, we evaluate the model on the validation set. After completing the optimization, we select the checkpoint that obtained the best validation results to evaluate on the testing set and report the result. We conduct our experiments on one Nvidia Telsa V100 32G GPU.

For `memdrop`, we tried different dropout probabilities from the range $[0.1, 0.2, 0.3, \cdots, 1.0]$. We finally used $0.7$ for dropout as it gives the highest performance on the validation set. In the diffusion prior, the number of diffusion steps during inference is set to 50. It was chosen from the range $[50, 100, 150, 200]$ following the same standard as described above. We set the variance schedule to constants increasing linearly from $\beta_1 = 5^{-6}$ to $\beta_M = 10^{-3}$. In the decoding process, for the beam search decoding method, the beam width is set as 5. For the sampling method, we set $K$ as 50 and $p$ as 0.9. All experiments are run only once due to resource constraints, with random seed set to 1234.

## B    Data statistics

Detailed dataset statistics can be seen in Tab. 6, where **#Examples** denotes the number of the dialog data example in the dataset, **#Turns** denotes the average number of turns, **#Tokens** means the average number of tokens in the dialog context and the response, respectively. We pre-process the multi-turn dialog to many single-turn dialogs as input to the model following DialogVED (Chen et al., 2022), where the dialog history is concatenated as a whole sequence with a special token [SEP] functioning as a separation mark to separate different turns. We use two special tokens, `madeupword01` and `madeupword02` from the BART model as the speaker indicator tokens. **#Train**, **#Valid**, and **#Test** denotes the number of the single-turn dialog in the training set, validation set and testing set of each dataset.

## C    Automatic Evaluation Metrics

In this section, we present the formal definitions of all metrics used in this paper. The BLEU-$n$ score is defined as

$$\text{BLEU-}n = \text{BP} \cdot \exp\left(\sum_{i=1}^{n} w_i \log \text{p}_i\right)$$
$$\text{BP} = \begin{cases} 1 & \text{if } c > r \\ e^{1-r/c} & \text{if } c \leq r \end{cases} \tag{17}$$

where BP is the brevity penalty term, $c$ denotes the length of the candidate generated response and $r$ denotes the effective reference corpus length, $w_i$ denotes the positive weights summing to one. $p_i$ is the geometric average of the modified $n$-gram precisions, which is defined as

$$p_n = \frac{\sum_{\mathcal{C} \in \{\text{Candidates}\}} \sum_{n\text{-gram} \in \mathcal{C}} \text{Count}_{\text{Clip}}(n\text{-gram})}{\sum_{\mathcal{C}' \in \{\text{Candidates}\}} \sum_{n\text{-gram}' \in \mathcal{C}'} \text{Count}(n\text{-gram}')} \tag{18}$$

where $\text{Count}(n\text{-gram}')$ denotes the the number of $n$-gram occurrences in the generated response, $\text{Count}_{\text{Clip}}(n\text{-gram})$ denotes the number of times the generated response's $n$-gram appears in the reference.

Distinct-$n$ score is calculated by countering the number of distinct unigrams and bigrams in generated responses. The value is scaled by total number of generated tokens to avoid favoring long sentences. Formally, it's defined as

$$\text{Distinct-1} = \frac{\text{Count}_{\text{Unique}}(1\text{-gram})}{\text{Count}(1\text{-gram})}$$
$$\text{Distinct-2} = \frac{\text{Count}_{\text{Unique}}(2\text{-gram})}{\text{Count}(2\text{-gram})}$$

where $\text{Count}_{\text{Unique}}(n\text{-gram})$ denotes the number of unique $n$-gram in the sentence. In this paper, we focus on Inter-Distinct score, namely the distinct score of generated responses in the whole test set.

The concept of "Entropy-$n$" is commonly used in information theory to quantify the average amount of information or uncertainty contained in a sequence of symbols of length "n". It measures the predictability or randomness of a sequence. Formally, it is defined as

$$\text{Entropy-}n = -\sum_{\omega \in \Omega} p(\omega) \log(p(\omega))$$

where $\Omega$ is the set of all the kinds of $n$-gram subsequence in a generated response. $p(\omega)$ denotes the normalized frequency of occurrence of the $n$-gram subsequence $\omega$.

| Dataset | #Examples | #Turns | #Tokens | #Tran | #Valid | #Test |
|---------|-----------|--------|---------|-------|--------|-------|
| DailyDialog | 76k | 5.9 | 75.6/15.0 | 76052 | 7069 | 6740 |
| Persona-Chat | 122k | 8.4 | 95.1/12.2 | 122499 | 14602 | 14056 |

Table 6: Data statistics of the datasets used in this paper

For model-based metrics, we can directly get the evaluation result from the evaluation model. The FBD metric is a fine-grained evaluation metric that can provide evaluation scores for 17 aspects. We only take the aspects that are the most relevant to this paper, including *Relevant*, *Correct*, *Coherent*, *Error Recovery*, *Consistent* and *Diverse*, and calculate an average to get the final FBD score.

## D  Model Comparisons

We compare Dior-CVAE with the state-of-the-art models for variational dialog generation:

- LIC (Golovanov et al., 2019): a PLM fine-tuned on the open-domain dialog datasets.
- ProphetNet (Qi et al., 2020): a PDG pre-trained on predicting more than one future tokens.
- DRESS (Han et al., 2022): a PLM fine-tuned to produce a balanced semantic distribution over the generated responses.

We also prepare and evaluate these baselines:

- iVAE$_{MI}$ (Fang et al., 2019): an implicit VAE model based on LSTMs that uses a NN to produce the posterior distribution.
- Optimus (Li et al., 2020): a pre-trained Transformer VAE for text generation.
- MVP+S (Tang et al., 2023): a multi-task supervised pre-trained model for text generation.
- DELLA (Hu et al., 2022b): the original model is a GPT-2-based HCVAE; we reproduced the model for the two evaluation datasets and replaced GPT-2 by BART for a fair comparison.

Additionally, we include results of the models taking advantage of large-scale dialog pre-training:

- PLATO (Bao et al., 2020): a large-scale pre-trained DRG model that uses a discrete latent variable to address the one-to-many problem.
- DialogVED (Chen et al., 2022): a Transformer VAE pre-trained on large-scale dialog data in order to improve DRG.

We use *-sample* to denote sampling from the top-$k$ tokens with the top-$p$ probabilities at each decoding step (Fan et al., 2018; Holtzman et al., 2020) and *-beam* to indicate beam search.

| Model | Dior-CVAE | DialogVED | DELLA(BART) |
|-------|-----------|-----------|-------------|
| #tok/s | 123.95 | 131.34 | 128.28 |

Table 7: Number of generated tokens per second (#tok/s) measured on the test set of `DailyDialog`.

## E  Human Evaluation

This section introduces the questions corresponding to four criteria used in our human evaluation. Each criterion are rated with a 3-point Likert scale.

- Coherence (COH): is the response relevant to the dialog?

- Informativeness (INF): does the response provide correct information?

- Safety (SAF): is the response safe to read?

- Engagement (ENG): do you want to have a conversation with this speaker?

The first three criteria are turn-based evaluation and the last one is evaluated on dialog-level.

## F  Inference Speed

Although there are concerns about the low speed of inference about the diffusion model especially in the synthetic image generation task. And there are many studies trying to improve the speed of inference of the diffusion model. While in this paper, the inference speed of our model should not be a major problem. This slow sampling nature of diffusion exists in synthetic image generation, where the number of diffusion steps is often set to $1000 - 4000$, and the dimension of the latent variables is relatively large (e.g., 128x128). In our method, the number of diffusion steps is set to 50 ($<$ 1,000) and the dimension of the latent variable is set to 64. The inference speed evaluated by generated tokens per second can be seen in the Tab. 7 We can see that the inference speed of Dior-CVAE doesn't drop significantly compared with the other two models.

| Context | A: I saw on the tv yesterday that there has been another earthquake in iran . |
| | B: Yes . There have been a few there recently . They say that this one was not a big quake . The Iranians are dealing with it on their own . They have purchased some special equipment to find people buried |
| | A: Does the newspaper say anything about casualties ? |
| step=1 | have died in the earthquake . The count is up to 10 right now . |
| Step=10 | , yes . But so far , less than 20 people have died . |
| Step=20 | yes . But so far , less than 20 people have died . |
| Step=30 | said 10 people have died . The rest are in hospital . |
| Step=40 | right now , less than 20 people have died . But several are in hospital . |
| Step=50 | right now , less than 20 people have died . But over 100 are in hospital . |

Table 8: Genrated responses by Dior-CVAE and DialogVED for the same dialog context.

| Dropout Rate | Cosine Similarity |
|---|---|
| 0.1 | 0.99 |
| 0.4 | 0.94 |
| 0.7 | 0.91 |
| 1.0 | 0.88 |

Table 9: Effect of the Memory Dropout method

## G Realization of the Diffusion Process

In this section, we show the effect of the latent variable at each diffusion step. Specifically, for a model setting where the diffusion steps is set to 50, we perform the denoising of 1, 20, 30, 40 and 50 steps, respectively and then input the denoised latent variable to the decoder to see the generation result. From the generated text we can see that as the denosing step increases, the generated response can become more relevant to the dialog context. One of the generation results can be seen in Tab. 8.

## H Effect of the Memory Dropout

Our proposed memory dropout method is used to alleviate the problem of posterior collapse. The direct result of the posterior collapse is that the latent variable has no effect on the text generation process. To further verify the effect of the memory dropout, we sample from the prior distribution 5 times for a given context and then perform the decoding process subsequently. We then obtain the embedding of 5 generated responses using the pretrained BERT

| Context | A: Excuse me , are you Mrs . Green from England ? |
| | B: Yes , I am . . |
| | A: I'm pleased to meet you , Mrs . Green . My name is Myra . I work in Beijing Institute of Technology . I came here to meet you . |
| Dior-CVAE | 1. How long will it take to get there ? |
| | 2. How long will it take to get there ? |
| | 3. This is our office block . We have all the administrative department here , Sales , Accounts , Personnel , Market Research and so on . |
| | 4. How many departments do you have ? |
| | 5. How can you work there ? |
| DialogVED | 1. Thank you for coming , ms . green . |
| | 2. Thank you very much . |
| | 3. Thank you very much . |
| | 4. Thank you very much . |
| | 5. Yes , i ' ve been working here for 3 years . |

Table 10: Genrated responses by Dior-CVAE and DialogVED for the same dialog context.

model and calculate the average cosine similarity between the responses. Experimental results on the test set of DailyDialog can be seen in the Tab. 9. An extremely high similarity means that all the five responses are almost the same no matter the how the sample latent variable changes. We can see that as the dropout rate increases, the similarity decreases, which can demonstrate the effect of the memory dropout method.

## I Case Study

We show more generation examples by Dior-CVAE and DialogVED on the test set of DailyDialog in this section. The dialog context and generated responses can be seen in Tab. 11. From the results, we can see that Dior-CVAE can generate more diverse responses than the SoTA model DialogVED.

| Context | A: Hello ! |
|---------|-----------|
| | B: Hello , Is that Stefan ? |
| | A: I'm sorry , but I'm afraid you've dialed the wrong number.There is no Stefan here . |
| Dior-CVAE | 1. Wrong number ? Are you sure ? |
| | 2. Sorry . |
| | 3. Nice to meet you . |
| | 4. Nice to meet you . |
| | 5. Don't hold on a second . I'm sure they've got it . |
| DialogVED | 1. Oh , I ' m sorry to bother you . |
| | 2. Oh , i ' m sorry to bother you . |
| | 3. Oh . are you sure ? |
| | 4. Oh , i ' m sorry to cause you a lot of trouble . |
| | 5. Oh , i ' m sorry to cause you a lot of trouble . |

| Context | A: Thank your for calling World Airline . What can I do for you ? |
|---------|-----------|
| | B: I need to book a plane ticket to London . |
| | A: Round-trip or one-way ? |
| Dior-CVAE | 1. It's flight CA169 , which leaves Beijing at 8: 00 a . m . on January 20th . |
| | 2. It's a flight tomorrow , and my name is turkey . |
| | 3. It's a flight tomorrow , and my name is turkey . |
| | 4. It's flight CA169 , which leaves Beijing at 8: 00 a . m . on January 20th . |
| | 5. I'd like to book a ticket . |
| DialogVED | 1. Round trip , please . |
| | 2. Round trip , please . |
| | 3. Round trip , please . |
| | 4. One way trip , please . |
| | 5. One way trip , please . |

| Context | A: What can I help you with ? |
|---------|-----------|
| | B: May I ask you something ? |
| | A: What's your question ? |
| Dior-CVAE | 1. I don't know how to say it in English . |
| | 2. I don't know how to say it in English . |
| | 3. I don't know about the position . |
| | 4. I don't know how to tell you . |
| | 5. I don't know how to tell you . |
| DialogVED | 1. Can you tell me how many credits would be enough for a bachelor ' s degree ? |
| | 2. Can you tell me how many credits would be enough for a bachelor ' s degree ? |
| | 3. Can you tell me how many credits would be enough for a bachelor ' s degree ? |
| | 4. Can you tell me how many credits would be enough for a bachelor ' s degree ? |
| | 5. Can you tell me how many credits would be enough for a bachelor ' s degree ? |

Table 11: Case study of response generation on DailyDialog

