# OpenReview forum: "Dior-CVAE: Pre-trained Language Models and Diffusion Priors for Variational Dialog Generation"
_EMNLP/2023/Conference — EMNLP 2023 Findings_

### Official Review · Reviewer_YFb3 · 2023-07-26

**Soundness:** 3

**Excitement:**

3: Ambivalent: It has merits (e.g., it reports state-of-the-art results, the idea is nice), but there are key weaknesses (e.g., it describes incremental work), and it can significantly benefit from another round of revision. However, I won't object to accepting it if my co-reviewers champion it.

**Paper Topic And Main Contributions:**

This paper focuses on two problems in dialog generation: 1) ignoring the latent variables, which is also known as the posterior collapse problem; 2) low diversity in responses due to the limited expressive power of Gaussian distributions. To tackle these two problems, the authors firstly use the memory dropout to tackle the posterior collapse. Then, a diffusion process is adopted to obtain a more complicated prior distribution. The experimental results demonstrate the effectiveness of their proposed method.

**Questions For The Authors:**

1. Are there any differences between the diffusion process adopted in the work and previous work? Do you meet any problems when drawing the method into dialog generation?

2. Could you give more explanations about memory dropout? What are the differences between Eq. 12 and Eq. 13? Do you dropout certain dimension of hidden vectors or directly mask some vectors directly? If you only dropout certain dimension of hidden vectors, is fig. 2 not accurate enough? If you mask some vectors directly, why does the decoder can generate fluent sentences when some of the hidden representations are completely lost? Why does memory dropout can tackle the posterior collapse problem? Are there any experiments which can demonstrate it directly? From Table 3, dropout can increase diversity by sacrificing fluency, but it does not seem to be the direct evidence to demonstrate that memory dropout can tackle the posterior collapse problem. Is it possible that the proposed model actually does not suffer from the posterior collapse problem? If it has the posterior collapse problem, why does the model can obtain close performance without memory dropout? Could you give more illustrations about it?


**Reasons To Accept:**

1. The motivation is clear and strong.

2. The effectiveness of the proposed model is demonstrated in the experiment.

**Reasons To Reject:**

1. Adopting diffusion process to model the latent space has been studied in previous work, and the authors directly draw the method into their models without any in-depth analyses.

2. The illustrations of the method is not clear to understand. The illustration of the memory dropout, one of the key proposed techniques in the paper, is not clear. And it is difficult for me to judge whether it is a reasonable method to tackle the posterior collapse problem.


**Reproducibility:**

3: Could reproduce the results with some difficulty. The settings of parameters are underspecified or subjectively determined; the training/evaluation data are not widely available.

**Reviewer Confidence:**

3: Pretty sure, but there's a chance I missed something. Although I have a good feel for this area in general, I did not carefully check the paper's details, e.g., the math, experimental design, or novelty.

**Typos Grammar Style And Presentation Improvements:**

The illustrations about the proposed method (especially the proposed techniques about the main contributions) should be clearer.

---

> ### Author Rebuttal · Authors · 2023-08-29
>
> We thank the reviewer for their thoughtful feedback. We are glad that you find that our motivation is clear and strong and the proposed method is effective and demonstrated in the experiments.
>
> We address the comments below.
>
> **1. “Adopting diffusion process to model the latent space has been studied in previous work, and the authors directly draw the method into their models without any in-depth analyses.”**
>
> - Answer: Adopting the diffusion model to build the prior distribution of VAE-based response generation model is not simple drawing and concatenating. The oversimplified Gaussian distribution assumption made by the VAE model for response generation is the main motivation for many studies on improving VAE models  (L86-96 and 558-573). Prior studies which build the prior distribution of a VAE model using the diffusion model are mostly in computer vision (CV,  L98). We describe the differences between this work and the CV work in the following point.
>
> **2. “differences between the diffusion process adopted in the work and previous work?”**
>
> - Answer: There are 2 key differences. We will emphasize the differences in the revision.
>     - (1) We introduce a series of interdependent latent variables from the encoder and the prior distribution of each latent variable is provided by a diffusion model. In other words, there is a series of diffusion models (L183 – 190). In contrast, the previous works consider only one single diffusion model (L589-593).
>     - (2) Previous works mainly focus on visual two-dimensional data (L574 – 577), which then often uses a convolutional neural network as the denoising network. In our work, the diffusion model is in contrast designed to process vector samples of the latent variables.  So we only use a fully-connected neural network as the denoising network.
>
>
> **3. “Do you meet any problems when drawing the method into dialog generation?”**
>
> - Answer: As explained in L328-337, the widely-adopted pre-trained encoder-decoder Transformer model has the cross-attention module that causes the main challenge when integrating Transformer into VAE model. The latent variables inferred from the encoder and the deterministic encoder output have some overlapping encoded information. The cross-attention module can provide the decoder a direct connection with the encoder output, then it can access to the information encoded by the encoder directly and neglect the latent variables, which can cause the posterior collapse problem. Additionally, the pre-trained Transformer decoder model can predict next words solely conditioned on the already generated text, which can further explain the posterior collapse problem (Bowman et al, L61).
>
> **4. “The illustrations of the method is not clear to understand. The illustration of the memory dropout, one of the key proposed techniques in the paper, is not clear.”**
>
> - **What are the differences between Eq. 12 and Eq. 13?**
>
>     - Answer: Thanks for pointing this out! This is an error in the writing of equation 13, which should be $\mathtt{XAtt}^l([\mathtt{Linear}(\mathbf{z}^t),\mathtt{memdrop}(\mathbf{H}_c^{\texttt{Enc}_L})])$. We have fixed this in the paper.
>
> - **Explanations about memory dropout**
>
>     - Answer: In general, we randomly mask out the contextualized token embeddings, i.e. the output of the encoder. Our method is a variant of the “word dropout” method (L60-74), where some input words of an RNN decoder are replaced by the unknown word tokens to weaken the decoder. Different from RNN, Transformer decoder has direct access to the encoder’s output through the cross-attention. We thus mask out part of the encoder output to enforce the decoder model relying more on the latent variables as described in L328-332.
>
> - **Why does the model obtain close performance without memory dropout?**
>
>     - Answer: The posterior collapse mainly refers to the decrease of the effects of the latent variables in the generation process, i.e. the VAE model may degenerate to the general AE model (Bowman et al, L61). Different from without pre-trained models, posterior collapse in this case results in fine-tuning the pre-trained model (BART) for dialogue generation and ignoring the latent variables. This consequently does not result in a significant decrease in generation performance.
>
> - **Why does memory dropout can tackle the posterior collapse problem?**
>
>     - Answer: Through masking out part of the encoder output, memory dropout then encourages the decoder to refer more to the latent variables rather than to rely on the encoder output through the cross-attention in BART.
>
> - **Demonstration of the posterior collapse without memory dropout**
>
>     - Answer: We trained and evaluated the ablation of our proposed model without the memory dropout module (Table 3 – row 3). The ablation generates the exact same response given the same context (using beam search) w.r.t. different sampled priors. The analysis will be included in the revision.

---

### Official Review · Reviewer_UuVZ · 2023-08-04

**Soundness:** 3

**Excitement:**

3: Ambivalent: It has merits (e.g., it reports state-of-the-art results, the idea is nice), but there are key weaknesses (e.g., it describes incremental work), and it can significantly benefit from another round of revision. However, I won't object to accepting it if my co-reviewers champion it.

**Missing References:**

https://arxiv.org/abs/1907.10568 (Investigating evaluation of open-domain dialogue systems with human generated multiple references)


**Paper Topic And Main Contributions:**

The paper proposes a hierarchical conditional variational autoencoder (CVAE) model with an informative prior produced by a diffusion model. The model is designed to address the posterior collapse problem and the Gaussian assumption made on the prior distribution, which restricts the diversity of generated responses. The paper derives a series of layer-wise latent variables from the encoder into the decoder as latent memory to alleviate the posterior collapse problem. The paper also proposes memory dropout to the cross-attention mechanism during training to encourage the use of latent variables for response generation. The prior distribution is parameterized with a diffusion model to increase the complexity and compatibility of the prior distribution. The paper shows that the proposed model outperforms existing response generation methods without relying on large-scale dialog pre-training.
Contribution:
1. The paper proposes a hierarchical CVAE model with an informative prior produced by a diffusion model to address the posterior collapse problem and the Gaussian assumption made on the prior distribution, which restricts the diversity of generated responses.
2. The paper derives a series of layer-wise latent variables from the encoder into the decoder as latent memory to alleviate the posterior collapse problem.
3. The paper proposes memory dropout to the cross-attention mechanism during training to encourage the use of latent variables for response generation.
4. The paper parameterizes the prior distribution with a diffusion model to increase the complexity and compatibility of the prior distribution.
5. The paper shows that the proposed model outperforms existing response generation methods without relying on large-scale dialog pre-training.

**Reasons To Accept:**

1. The paper proposes a novel approach to address the posterior collapse problem and the Gaussian assumption made on the prior distribution, which restricts the diversity of generated responses.
2. The paper shows that the proposed model outperforms some response generation methods (like Dialog VED) without relying on large-scale dialog pre-training.
3. The paper provides a comprehensive evaluation of the proposed model on two commonly-used open-domain dialog datasets.
4. The paper proposes memory dropout to the cross-attention mechanism during training to encourage the use of latent variables for response generation.

**Reasons To Reject:**

This paper has a sound idea to use diffusion model to model the complex prior. However, the main limitation of this paper is outdated method to evaluate dialog systems in the era of ChatGPT. It is reasonable to evaluate dialog response generation models using BLEU, Distinct-1/2 a few years ago. However, there are many research that have already shown limitations of these shallow metrics.

Thus, current experiment results is able to show the proposed DiorCVAE is able to outperform DialogVED, but not sufficient enough to show:
1. the proposed method is able to actually generate fluent conversation with diverse human users.
2. the learned diffusion prior is able to learn the complex prior distribution (as shown in Figure 1)
3. Can this proposed method be extended to larger-scale training or LLM.

**Reproducibility:**

4: Could mostly reproduce the results, but there may be some variation because of sample variance or minor variations in their interpretation of the protocol or method.

**Reviewer Confidence:**

4: Quite sure. I tried to check the important points carefully. It's unlikely, though conceivable, that I missed something that should affect my ratings.

---

> ### Author Rebuttal · Authors · 2023-08-29
>
> We thank the reviewer for their positive feedback and insightful comments. We are really encouraged that the reviewer found that our work proposes a novel approach to address posterior collapse and the simplified Gaussian assumption for variational response generation models.
> We address the concerns raised by the reviewer below.
>
>
> 1. **“The main limitation of this paper is outdated method to evaluate dialog systems in the era of ChatGPT and the missing reference to evaluation proposed by Gupta et al., (2019)”**
>     - Answer:Answer: We partially disagree with the reviewer. Although several metrics based on pre-trained models have been proposed, BLEU and Distinct-½ metrics (used in our evaluation) remain to be the must-have metrics for the evaluation of response generation, even in ACL2023 [1,2,3].
> While Gupta et al., (2019) emphasized on a multi-reference test set, we note that a multi-reference dataset (DailyDialog++) is indeed used in our human evaluation (L520-524). Human evaluation is still the must-have evaluation for response generation.
> Nevertheless, we will consider the recent published metrics in ACL 2023 such as LLM-EVAL (Lin & Chen, NLP4ConvAI 2023).
>
> 2. **“whether the proposed method is able to actually generate fluent conversation with diverse human users.”**
>     - Answer: Reviewer’s suggestion is exactly what we want to show with our human evaluation, which focuses on evaluating the ability of our model for generating diverse responses conditioned on the same dialogue context.
>
> 3. **“whether the learned diffusion prior is able to learn the complex prior distribution (as shown in Figure 1)”**
>     - Answer: Thanks for the suggested analysis! We will include the multimodal analysis in the revision. We conduct the analysis by sampling 5-10 data samples in the test set and sampling 500 points from the learned diffusion prior distribution given the same dialogue context of the data sample and visualizing them to verify whether our model learns a more complex distribution. We are happy to also include more analysis if the reviewer has other concrete suggestions.
>
> 4. **“Can this proposed method be extended to larger-scale training or LLM.”**
>     - Answer: Our method can be extended to larger-scale training or LLM. Pretrained language model with encoder-decoder architecture can be easily used as a substitution of BART in our method. Besides, we can also adapt a pre-trained language model with decoder-only architecture like GPT-2 to the VAE model following Hu et al., (2022).
> However, fine-tuning LLMs requires a significant amount of computational resources, future work can investigate using parameter efficient finetuning (PEFT) methods while turning LLMs to VAEs with diffusion priors. Since the investigation of using PEFT is not in the scope of this work, we will leave it for future work.
>
>
>
>
>
> [1] [PVGRU: Generating Diverse and Relevant Dialogue Responses via Pseudo-Variational Mechanism](https://aclanthology.org/2023.acl-long.185) (Liu et al., ACL 2023)
> [2] [Enhancing Personalized Dialogue Generation with Contrastive Latent Variables: Combining Sparse and Dense Persona](https://aclanthology.org/2023.acl-long.299) (Tang et al., ACL 2023)
> [3] [Envisioning Future from the Past: Hierarchical Duality Learning for Multi-Turn Dialogue Generation](https://aclanthology.org/2023.acl-long.407) (Lv et al., ACL 2023)

---

### Official Review · Reviewer_xnUx · 2023-08-07

**Soundness:** 2

**Excitement:**

2: Mediocre: This paper makes marginal contributions (vs non-contemporaneous work), so I would rather not see it in the conference.

**Paper Topic And Main Contributions:**

This paper integrates Gaussian diffusion into a dialogue generation framework, specifically the CVAE model, to tackle the issue of isotropic Gaussian priors in VAEs that can result in posterior collapse. The method also incorporates dropout on memory latents to mitigate the collapse problem. The outcomes of this approach demonstrate competitive results on open-domain dialogue generation datasets.

**Reasons To Accept:**

The motivation behind the research is clear and straightforward. While there is no explicit confirmation of the multimodality of the priors generated by diffusion models in Figure 1, the proposed method demonstrates measurable enhancements in overall performance compared to baseline models.

**Reasons To Reject:**

The primary contribution of this conference paper centers on the adoption of a diffusion model as the prior generator. However, my principal concern revolves around the experimental validation of this proposition.

The clarity of the proposed method's contribution remains obscured. The paper allocates only a limited portion of its main section to delineate the designs and procedures of the diffusion prior. Consequently, the holistic integration of the diffusion process with CVAE training lacks elucidation. Specifically, the utilization of the sampled posterior $q(z|r,c)$ in Equation 2 as the original data $x_0$ for diffusion is addressed, yet there is an absence of explanation concerning the dynamics governing the evolution of $z$ and whether $z$ is subject to change at each time step. Given the potential instability of training both diffusion models and VAE, a comprehensive insight into the co-training process of diffusion latents with CVAE is imperative. Additionally, an exposition on the parameter tuning process is warranted to enhance the readers' understanding of the intricate interplay.

Considering the inherently slow sampling nature of diffusion, an essential facet to address would be a comparative analysis of inference speed against relevant baselines. Furthermore, it is recommended to include a comparison of parameter sizes between the proposed model and the baseline models mentioned in Table 1. While the paper references baselines like PLATO, it is pertinent to consider more recent alternatives such as [1], as they could offer a more current benchmark for evaluation.

[1] Chen, Ruijun, Jin Wang, Liang-Chih Yu, and Xuejie Zhang. "Learning to Memorize Entailment and Discourse Relations for Persona-Consistent Dialogues." arXiv preprint arXiv:2301.04871 (2023).

=============

I have carefully read the author's response. However, most of my concerns are not well addressed. The paper's contribution is not clearly explained, with only a limited section dedicated to describing the diffusion prior. I suggest the authors revise and resubmit.

**Reproducibility:**

2: Would be hard pressed to reproduce the results. The contribution depends on data that are simply not available outside the author's institution or consortium; not enough details are provided.

**Reviewer Confidence:**

4: Quite sure. I tried to check the important points carefully. It's unlikely, though conceivable, that I missed something that should affect my ratings.

---

> ### Author Rebuttal · Authors · 2023-08-29
>
> We thank the reviewer for their thoughtful feedback. We are encouraged that they find that our motivation behind the research is clear and straightforward, the proposed method demonstrates measurable enhancements in overall performance compared to baseline models.
> We address reviewer comments below.
>
> 1. **“There is no explicit confirmation of the multimodality of the priors generated by diffusion models in Figure 1.”**
>     - Answer: Thanks for the suggested analysis! We will include the multimodal analysis in the revision. We conduct the analysis by sampling 5-10 data samples in the test set and sampling 500 points from the learned diffusion prior distribution given the same dialogue context of the data sample and visualizing them to verify whether the learned distribution is more complex. We are happy to also include more analysis if the reviewer has other concrete suggestions.
>
> 2. **“The paper allocates only a limited portion of its main section to delineate the designs and procedures of the diffusion prior.”**
>
>     - **there is an absence of explanation concerning the dynamics governing the evolution of ​z and whether z​ is subject to change at each time step.**
>         - Answer: Section 2.4 presents the reverse diffusion process, where Eq. 5 is the key formula that controls how z changes at each time step. We will show the generated responses conditioned on z sampled from each of the step to demonstrate how the z changes in revision.
>
>      - **comprehensive insight into the co-training process of diffusion latents with CVAE is imperative. an exposition on the parameter tuning process is warranted to enhance the readers' understanding of the intricate interplay.**
>         - Answer: We will visualize the changes of the loss function in the revision. We describe detailedly the training and inference process of our model in the Figure 3 and Figure 4 of the appendix F, which can offer an exposition on how we train the model and make inference in practice.
>
>
> 3. **an essential facet to address would be a comparative analysis of inference speed against relevant baselines.**
> Answer: In our experiments, the average inference speed of our model is 1.37s per batch, and DialogVED and DELLA(BART) are 1.31s per batch and 1.21s per batch, respectively (batchsize=4, see Appendix). We will put the analysis in the revision. The inference speed of our model should not be a major problem. This slow sampling nature of diffusion exists in synthetic image generation, where the number of diffusion steps is often set as 1000 – 4000, and the dimension of the latent variables is relatively large (e.g., 128x128). In our method, the number of diffusion steps is set as 50 (< 1,000) and the dimension of the latent variable is set as 64 (see Appendix).
>
>
> 4. **“it is recommended to include a comparison of parameter sizes between the proposed model and the baseline models mentioned in Table 1.”**
>     - Answer: The rightmost column of Table 1 shows exactly the parameter sizes of all models in comparison. We will recap this column in the caption and add the corresponding text for clarification.
>
>
> 5. **While the paper references baselines like PLATO, it is pertinent to consider more recent alternatives such as Chen et al. 2023, as they could offer a more current benchmark for evaluation.**
>     - Answer: We respectfully disagree. Our baselines include the latest models from 2022: DRESS, MVP+S, DELLA and DialogVED.
> The most important comparison is DialogVED, which is also considered in Chen et al. (2023). While we agree that the paper can be cited in the related work, we emphasize 3 main reasons for not including their work in the main comparison as follows. First, they focus more on the persona-consistency of the generated responses rather than diversity, which is the main focus of our work. Consequently, they reported results on different datasets compared to ours and the reported results on the DSTC7-AVSD dataset are mostly lower than the DialogVED (all BLEU scores are lower). Last, their method relies on finding a suitable additional NLI dataset w.r.t. each dialog dataset.
> Without the NLI component and additional training objectives (bag-of-word prediction, dialogue discourse memory and language modeling), their model reduces to our reported BART ablation (Table 3, Row 4) but with only one latent variable. While our reported ablation is equipped with a series of latent variables, we will include the ablation model with only one latent variable in the revision.
>
>
> 6. **Reproducibility**
>     - The experimental settings can be seen in Appendix. We also include the source code in the supplemental material in the submission for reproducibility.

---

### Meta-Review · Area_Chair_6kgG · 2023-09-18

**Recommendation:** 3

**Metareview:**

Based on the reviews and updating scores provided, the paper titled "Gaussian Diffusion into a Dialogue Generation Framework” primarily showcases integration of Gaussian diffusion and dropout on memory latents within a dialogue generation framework, specifically the CVAE model, to alleviate the impact of isotropic Gaussian priors in VAEs and their resulting posterior collapse. This approach shows promising results on open-domain dialogue generation datasets.

## Pros:

- Proposes a novel approach to handle the posterior collapse problem and the limitation of Gaussian distribution on prior, which could enhance the diversity of generated responses.
- The proposed model shows potential competitive results against some existing generation methods without relying on large-scale dialog pre-training.
- There is a clear and strong motivation behind the research, with the results demonstrating effectiveness of the proposed model.

## Cons:

- One of the main criticisms revolves around the clarity and thoroughness of the proposed method's explanation. Detailed descriptions and insights into the co-training process of diffusion latents with CVAE are missing, blurring potential readers' understanding.
- Integration of the diffusion process with the CVAE training is not clearly explained.
- Information is lacking on the dynamics governing the diffusion process and the parameter tuning process.
- There is a lack of comparative analysis on inference speed and parameter size against baselines.
- Experimental validation of the paper's claims could be more robust, with the inclusing of more recent baseline models for evaluation.
- Evaluation methods of the system are deemed outdated and insufficient, and fail to show if the model can actually generate fluent, diverse dialogues.
- As per the third reviewer, more in-depth analysis on the influence of the diffusion process on the latent space is needed.
- There are detailed concerns about the memory dropout method and questions on its effectiveness in regards to the posterior collapse problem.

---

### Decision · Program_Chairs · 2023-10-07

**Decision:**

Accept-Findings

**Comment:**

Based on the reviews and updating scores provided, the paper titled "Gaussian Diffusion into a Dialogue Generation Framework” primarily showcases integration of Gaussian diffusion and dropout on memory latents within a dialogue generation framework, specifically the CVAE model, to alleviate the impact of isotropic Gaussian priors in VAEs and their resulting posterior collapse. This approach shows promising results on open-domain dialogue generation datasets.

## Pros:

- Proposes a novel approach to handle the posterior collapse problem and the limitation of Gaussian distribution on prior, which could enhance the diversity of generated responses.
- The proposed model shows potential competitive results against some existing generation methods without relying on large-scale dialog pre-training.
- There is a clear and strong motivation behind the research, with the results demonstrating effectiveness of the proposed model.

## Cons:

- One of the main criticisms revolves around the clarity and thoroughness of the proposed method's explanation. Detailed descriptions and insights into the co-training process of diffusion latents with CVAE are missing, blurring potential readers' understanding.
- Integration of the diffusion process with the CVAE training is not clearly explained.
- Information is lacking on the dynamics governing the diffusion process and the parameter tuning process.
- There is a lack of comparative analysis on inference speed and parameter size against baselines.
- Experimental validation of the paper's claims could be more robust, with the inclusing of more recent baseline models for evaluation.
- Evaluation methods of the system are deemed outdated and insufficient, and fail to show if the model can actually generate fluent, diverse dialogues.
- As per the third reviewer, more in-depth analysis on the influence of the diffusion process on the latent space is needed.
- There are detailed concerns about the memory dropout method and questions on its effectiveness in regards to the posterior collapse problem.